# Anisotropy reveals contact sliding and aging as a cause of post-seismic velocity changes

Manuel Asnar [1,2] ✉, Christoph Sens-Schönfelder [1], Audrey Bonnelye[3], Andrew Curtis [4], Georg Dresen [1,5] & Marco Bohnhoff [1,2]

Rocks exhibit astonishing time-dependent mechanical properties, like memory of experienced stress or *slow dynamics*, a transient recovery of stiffness after a softening induced by almost any type of loading. This softening and transient recovery is observed in the subsurface and in buildings after earthquakes, or in laboratory samples. Here, we investigate the anisotropy of non-linear elastic effects in a sandstone sample under uniaxial loading. We report that slow dynamics is observed independently of propagation direction, while the acoustoelastic effect shows the expected anisotropy originating from the opening and closing of cracks. From this, we argue that slow dynamics is caused by the sliding of oblique grain-to-grain contacts and the resulting changes in frictional properties, as empirically described by rate-and-state friction and observed in laboratory experiments across block contacts. We establish a connection between the nonclassical nonlinearity of heterogeneous materials and the framework of rate-and-state friction, providing an explanation for the elusive origin of slow dynamics, and adding a different perspective for monitoring very early stages of material failure when deformation is still distributed in the bulk and begins to coalesce towards a fracture.

Material failure is not instantaneous, but rather results from the process of stress transfer through internal deformation. This transfer progressively localizes deformation on developing fractures and fault structures, concentrating the stress on asperities, until shear stress finally exceeds material strength[1]. Understanding and capturing these processes is paramount to the mitigation of natural and engineering hazards related to failure.

Depending on how far along it is, this process of stress transfer and concentration can be observed through changes in stiffness, macroscopic deformation, or acoustic emissions. In heterogeneous materials like rocks, concrete, or cracked materials, this process is governed by some notably nonlinear, time-dependent mechanical properties like hysteresis[2], a fading memory of the maximum experienced stress (e.g., Kaiser effect[3]), or the so-called *slow dynamics* (SD). This latter term, originally coined by the acoustics community[4], describes a transient process of stiffness recovery after a softening that

is induced by almost any type of static or dynamic mechanical or thermal perturbation. Such softening and transient recovery is observed in the subsurface[5–8], in buildings after earthquake shaking[9,10], in laboratory samples[11] made of rocks[4,12–14], concrete[15], packs of glass and aluminum beads[16,17] and even single bead contacts[18], and under both dry and fluid-saturated conditions[19]. Sudden changes in temperature can also trigger softening and SD[13,20], both when heating or cooling the sample. Remarkably, the SD response to a mechanical or thermal perturbation is symmetry-breaking; static loading and unloading as well as heating and cooling always cause a sharp softening followed by transient hardening, and never a sharp hardening followed by transient softening. A hallmark of the SD recovery is the shape of its temporal evolution, which is linear with respect to the logarithm of time elapsed since the end of the perturbation[4,11]. This is why the phenomenon is commonly referred to as *slow*, as opposed to the *fast* dynamics of the perturbation. In a laboratory setting, SD has

[1]GFZ Helmholtz Centre for Geosciences, Potsdam, Germany. [2]Department of Earth Sciences, Freie Universität Berlin, Berlin, Germany. [3]GeoRessources Laboratory, Université de Lorraine, Nancy, France. [4]School of GeoSciences, University of Edinburgh, Edinburgh, UK. [5]Institute of Geosciences, Universität Potsdam, Potsdam, Germany. ✉e-mail: masnar@gfz.de

been most apparent for fairly low confining pressures (0.1 MPa to 10 MPa) and small applied strains ($10^{-4}$ to $10^{-7}$) before vanishing at higher pressures[21]. The physical origin of this ubiquitous behavior remains enigmatic despite decades of research[22,23].

Although SD is linked to nonlinearity in the elastic properties[11], the transient softening and recovery cannot be reproduced in the frame of classical nonlinear elasticity, sometimes referred to as acoustoelasticity or finite strain theory[24,25]. The presumed existence of two different types of nonlinear elastic behavior in rocks has been mentioned in past studies[21,26,27]. Therefore, in contrast to the instantaneous *classical nonlinear* effects, we refer to processes involving a time dependency—such as SD—as *nonclassical* effects.

Models have been proposed attributing SD to the adsorption of fluid at the contacts between grains[28,29], the destruction and recreation of bonds in contact asperities[30], or more generally to internal damage processes[31,32]. The ubiquity of SD in vastly different experimental settings suggests a form of universality in the physics involved, regarding both the loss of stiffness and the $\log(t)$ recovery processes[20]. Neutron diffraction experiments have shown that the nonlinear and transient effects were concentrated at the grain boundaries[33], suggesting that the crux lies in defining what happens at the interface between grains or beads—in the so-called *bond system*[22].

Previous attempts at investigating the physical processes in this bond system focused on the logarithmic time dependency of the SD recovery in resonance experiments[34] and in localized measurements[19,35]. However, the omnipresence of this $\log(t)$ dependence in a broad range of physical phenomena[36] limits its informative value for the identification of the physical mechanisms from which SD originates in particular materials[37].

An alternative route to investigate the physical origin of the transient effects is the study of anisotropy—or lack thereof—which has only rarely been addressed. Using Dynamic Acousto-Elastic Testing experiments along the axial and radial directions in a limestone sample, a difference in the amplitudes of the dynamic classical nonlinear effect has been observed[12], but the recovery process was not investigated specifically, which limits the ability to separate instantaneous and transient effects. A sample being excited at the first longitudinal and torsional modes showed that the amplitude of the softening and recovery depends on the polarization of the probe wave[38]. Additionally, that study investigated the sample during the dynamic perturbation phase only, and without analyzing the relaxation phase that starts once the perturbation stops, hindering the ability to investigate the transient effect separately. The effect of crack orientations in anisotropic sandstones on nonlinear elastic effects has also been investigated[39]. However, no distinction between classical and nonclassical effects was made in that study, and only the conditioning phase was investigated. Therefore, despite observing a significant dependence of the amplitude of nonlinear effects on crack orientation in the sample, one cannot draw conclusions about the transient effect alone.

A lack of anisotropy of SD has been observed *en passant* in limestone[14] and in granite[40] at comparatively high pressures of 80 MPa, when acoustics experiments rarely go beyond 10 MPa. Friction and slip along grain boundaries and cracks was invoked to explain the transient effects[40]. While these experiments were performed with confining pressure ranges about 10 times larger than the upper limit mentioned above, the applied strains also significantly exceed the usual strains in acoustic experiments ($10^{-4}$ to $10^{-7}$) and field observations by two to five orders of magnitude. Finally, the differences in the softening and recovery signals between waves of three different polarizations have been investigated[41]. In summary, none of these studies provided an explicit analysis of the anisotropy of the softening and time-dependent SD recovery in the low-strain regime.

In this work, we investigate the anisotropy of both the instantaneous classical nonlinearity and the transient nonclassical nonlinearity using the relative seismic velocity changes of high-frequency, low amplitude compressional probe waves as a proxy for the dynamic modulus of the rock sample. We separate the two effects and independently measure their anisotropy to infer the physical mechanisms and structures from which each originates. We show that the respective anisotropies of the classical and nonclassical effects are irreconcilable, and that they must therefore originate from different processes or structures. From this, we propose that SD originates from the combination of a loss in frictional strength due to the sliding of tilted contacts between grains, followed by a healing phase arising from the transient increase in frictional strength during hold time, a phenomenon commonly known as *contact aging*.

## Results and discussion

### Classical and nonclassical nonlinearity

We submit a Bentheim sandstone sample that has a uniform grain structure, no significant structural anisotropy, and consists of virtually pure quartz grains[42–44] to a loading protocol with steps of static, vertical uniaxial compression (red background in Fig. 1), where the sample is held at a fixed strain of $\varepsilon = 5 \times 10^{-5}$. Relative velocity changes $\eta$ with respect to the start of the experiment are observed on paths that cross the sample at different angles $\theta$ with the loading axis. $\eta(t; \theta)$ is decomposed into canonical components, as described in Equation (3), reflecting the classical effect $C(t)$, which is proportional to the applied deformation, and a nonclassical effect $R(t)$ that features repeated damage-recovery sequences triggered by changes in the external deformation (Fig. 1).

Contributions of $C(t)$ and $R(t)$ to the velocity changes $\eta$ observed for a particular propagation path are estimated in terms of the coefficients $\beta$ and $\delta$ that scale the classical and nonclassical components $C(t)$ and $R(t)$ to fit the observed velocity changes.

### Anisotropy of the classical and nonclassical effects

Our measurements are based on relative velocity changes, meaning that any potentially present initial anisotropy of the velocity only has a second order influence on these measurements. This allows us to exclusively investigate the additional anisotropy induced by the loading during the experiment.

The governing parameter for variations in the shape of observed velocity changes is the aforementioned angle $\theta$. We quantify the anisotropy by fitting the angular dependence of $\beta(\theta)$ and $\delta(\theta)$ using:

$$\begin{aligned} \beta(\theta) &= u_C \cos^2(\theta) + v_C \\ \delta(\theta) &= u_{NC} \cos^2(\theta) + v_{NC}. \end{aligned} \qquad (1)$$

The coefficients $u$ and $v$ of this transverse isotropy law[45] parameterize the velocity change of the classical ($C$) and the nonclassical ($NC$) effects for propagation perpendicular to the loading axis ($v_C$ and $v_{NC}$), and their angular dependence ($u_C$ and $u_{NC}$). We discuss in detail the choice of this particular anisotropy law in the Supplementary Discussions 4 and 5. Figure 2 shows the angular dependence of $\beta$ and $\delta$ representing the classical and nonclassical components together with the corresponding parameters $u$ and $v$ of the anisotropy law fitted with 5000-iteration bootstrap sampling[46].

For the classical effect $\beta$, we obtain anisotropy coefficients with 1-$\sigma$ uncertainty: $u_C = (4.1 \pm 0.4) \times 10^{-2}\%$ and $v_C = (0.5 \pm 0.1) \times 10^{-2}\%$. For the nonclassical effect $\delta$ on the other hand, we have $u_{NC} = (0.7 \pm 0.4) \times 10^{-2}\%$ and $v_{NC} = (2.3 \pm 0.1) \times 10^{-2}\%$. Both $u_C$ and $u_{NC}$ have similar absolute uncertainty, as do $v_C$ and $v_{NC}$ (see Fig. 2).

Absolute values of $u$ and $v$ are very different for the classical and nonclassical effects. While the classical effect decribed by $\beta(\theta)$ is clearly anisotropic ($u_C/v_C > 1$), the nonclassical effect described by parameter $\delta(\theta)$ is significant in all directions ($v_{NC} > 0$) but only weakly anisotropic ($u_{NC}/v_{NC} < 1$).

These results show that the anisotropies of $\beta$ and $\delta$ are irreconcilable. Therefore, the classical and nonclassical effects cannot be caused by the same processes happening on the same microstructures. This does not depend on the particular model used to quantify the anisotropy (see Supplementary Discussion 4).

Together with the properties implemented in the shape of the components $C(t)$ and $R(t)$, we can summarize the respective characteristics of the classical and nonclassical effects. The classical effect: (i) depends on the sign of stress change; (ii) is significantly anisotropic and (almost) vanishes along the horizontal; (iii) is instantaneous, i.e., is correlated to the applied strain. In contrast the nonclassical effect: (i) is identical for both positive and negative changes of strain; (ii) is significant along all ray paths with almost no anisotropy; (iii) is not

correlated to the applied strain but is triggered by changes in applied strain, and relaxes as $\log(t)$.

## Disparity of structures or processes

Bentheim sandstone has been chosen for this experiment because of its simple structure consisting of relatively uniform quartz grains and cementation[44,47]. As indicated by neutron diffraction experiments, the deformation of sandstones in the low strain regime is predominantly (~80%) accommodated by a small fraction of the sample volume, namely the regions around grain contacts[33]. These regions form the so-called *soft bond system* that connects the stiff grain matrices. The bond system undergoes strong deformation and hosts the processes responsible for hysteresis and temporal modulus changes observed in

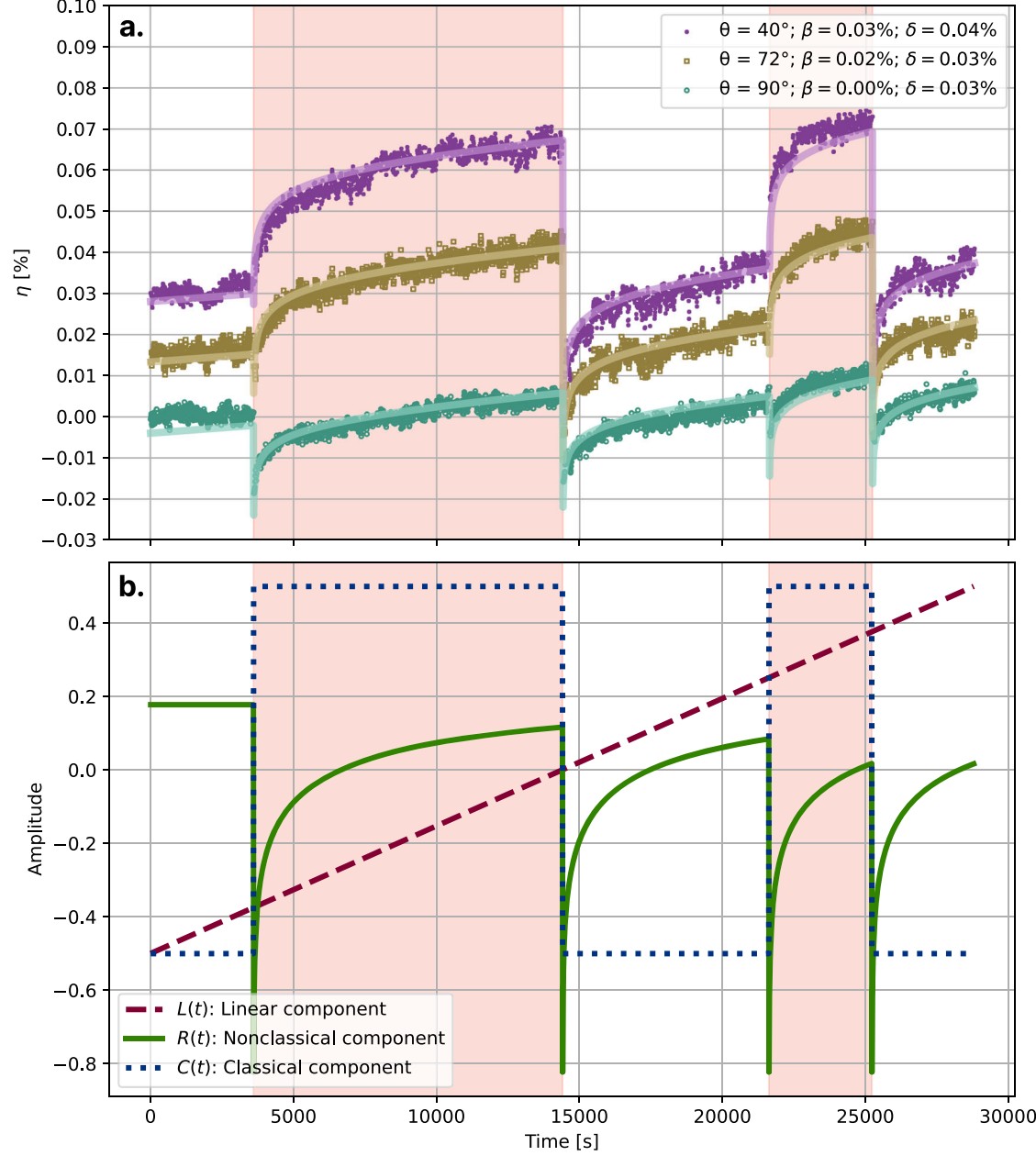

**Fig. 1 | Relative velocity change measurements and model components.**
**a** Relative velocity change measurements $\eta$ for three different values of $\theta$ (purple dots, brown squares and blue circles). Solid lines show the respective model fits: $\eta(t) \approx \alpha L(t) + \beta C(t) + \delta R(t)$. The colored background corresponds to periods where uniaxial strain is applied on the sample. The origin of each set of measurements has been offset by 0.015% for clarity. **b** The linear (dashed), nonclassical (solid) and classical (dotted) components for the linear inversion as described in Equation (3). The components are of unit amplitude, so that the corresponding coefficients $\alpha$, $\beta$ and $\delta$ can be compared. The mean of each component is subtracted prior to the linear inversion.

sandstones with similar composition and porosity to our sample[33]. The nonclassical component thus originates in the bond system[11]. Furthermore, the macroscopic modulus and acoustic wavespeed of sandstone is significantly lower and the nonlinearity significantly higher than the corresponding properties of the quartz minerals of our sandstone[48], indicating that the classical effect also chiefly originates from processes occurring in the soft bond system.

Therefore, in agreement with robust past evidence[22,33,49,50], we assume that both the classical and nonclassical components that we use to describe the observed velocity changes at the small stress levels used here ($\sigma_{11} \approx 50$ kPa) are caused by processes in the soft bond system in a sample that is macroscopically isotropic and elastic. However, as established above, the observed anisotropies point to fundamental differences of processes or structures at the root of the classical and nonclassical effects.

The fact that the nonclassical effect does not exhibit anisotropy while the classical effect does display the expected load-induced anisotropy (see Supplementary Discussion 5) has two fundamental implications. First, the two effects cannot be caused by the same processes on the same structures, e.g., opening and closing of the same cracks or contacts. Second, the microscopic processes responsible for the nonclassical response must either have an isotropic effect on velocity, or the affected structures must have a certain angular distribution for the anisotropy to vanish macroscopically.

A number of past studies have already hinted at there being two fundamentally distinct types of nonlinear elastic behavior[21,26,27,39], which often appear together in experiments. While it is commonly agreed that the opening and closing of cracks is responsible for a significant portion of the nonlinearity, it is unclear how it can generate two distinct behaviors by itself. As a result, many experimental studies are left with part of their observations being inconsistent with this explanation: for example, higher-frequency fluctuations in an otherwise low-frequency nonlinear response[39], or a single pump frequency-independent harmonic component in the nonlinear response[21]. Moreover, the $\log(t)$ recovery of the nonclassical component can only be explained by an additional time-dependent process, which exclusively affects the structures responsible for the nonclassical effect, as opposed to the classical effects which are time-independent[21].

## Real contact area and acoustic velocity

We suppose that the major process by which the soft bond system affects wave velocities is through changes in the real contact area between grains. The size distribution of the individual asperities forming the real contact area between various materials is fractal within the optical observational range[51]. The linear increase of the real contact area with applied load[51] is governed by the indentation yield strength, and occurs through an increase in both the number and size of real contacts, thereby maintaining the self-similarity of the size distribution. This indicates that there is no particular stress scale at which the behavior changes.

P-wave transmission through an interface between two blocks is sensitive to the real contact area[52,53]. Similarly, wave velocities sense the evolution of real contact area in a rock joint[54,55]. Although the setup of the block experiments with grain-to-grain contacts arranged along the block interface differs from our situation, it indicates that our velocity changes measured in the intact sample sense the real contact area of the asperities in weak contacts distributed in the bulk.

Concerning the classical effect, and based on these observations, we follow the commonly shared explanation[11,26,48,56] that it is mostly caused by the closing (opening) of compliant contacts accompanied by an increase (decrease) in the real contact area under compression (extension). Such opening and closing is consistent with the three

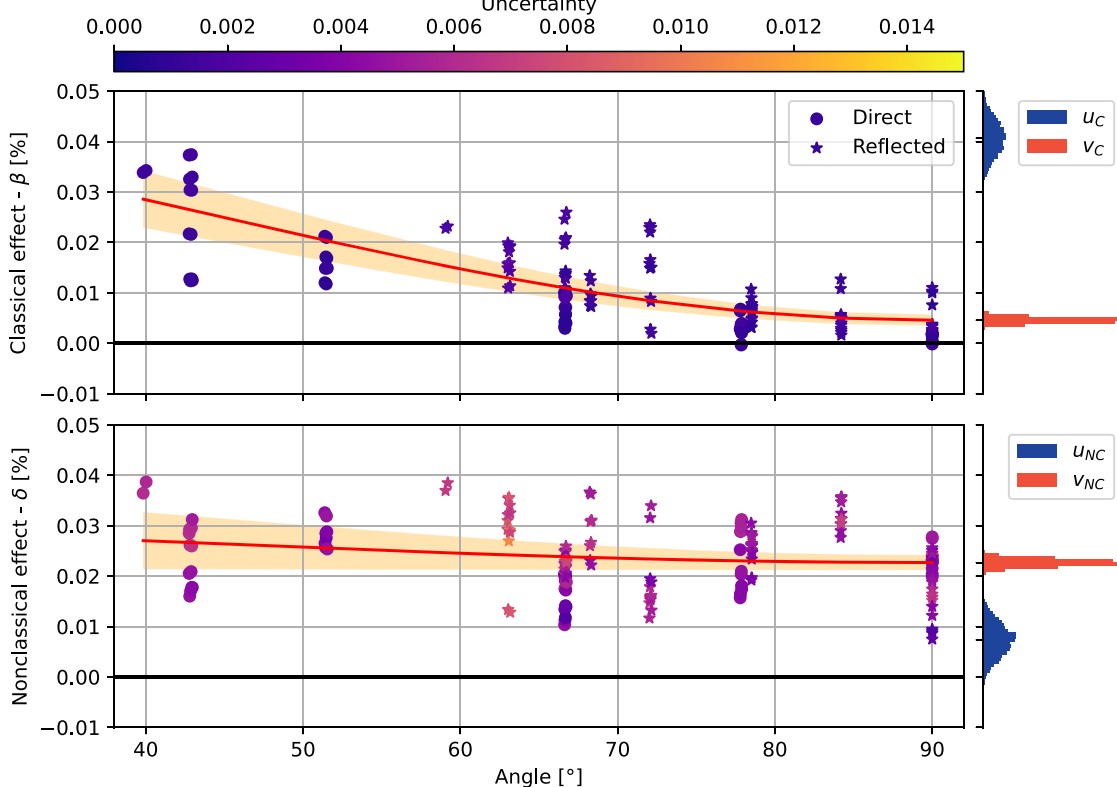

**Fig. 2 | Anisotropy of the classical and nonclassical effects.** Results of the fitting of the classical $\beta$ and nonclassical $\delta$ parameters to the transverse isotropy law in Equation (1). Dots indicate measurements resulting from direct P-wave arrivals; stars are for reflected P-wave arrivals. Color of symbols shows uncertainty of individual measurements. The red curve is the average model with orange bands indicating the 95% confidence interval. The histograms show the results of the 5000-iteration bootstrap sampling inversion of the $u$ and $v$ coefficients.

observations about the classical effect mentioned above: (i) its correlation with strain; (ii) its anisotropy[57]; (iii) its instantaneous response.

## Contact sliding as the origin of slow dynamics

In contrast to the classical effect, a change in real contact area induced by compression (extension) does not qualitatively match our observations of the nonclassical effect. This remains true even when including additional transient effects that might be associated with the opening and closing (e.g., indentation creep). Opening and closing of compliant cracks is ruled out by the clear difference in the observed anisotropies. Finally, the independence of the nonclassical effect on the sign of the strain change is fundamentally inconsistent with the mechanism described above.

Crucially, in Bentheim sandstone, the only available structures in the bond system are quartz grain-to-grain contacts, including potential cementation. Therefore, since the affected structures are of the same type, the difference between classical and nonclassical effects must be related to the processes occurring at these contacts.

We suggest that the main driver of the nonclassical effect is the sliding of soft contacts along planes which have to be inclined with respect to the principal stresses to experience shear stress. This idea has been the subject of extensive speculation already[20,26,30,58,59], but has been backed by little to no experimental evidence, until now. Figure 3 provides an illustration of both effects. The interpretation of the nonclassical effect as a result of contact sliding explains the instantaneous drop in velocities and the subsequent $\log(t)$ recovery (iii), as we show below. It also explains the identical response to positive and negative stress changes (i), as sliding along oblique planes occurs during both axial compression and decompression. However, what is the relation between sliding of contacts and the observed changes in acoustic velocity?

Again, we conjecture that the mechanism connecting contact slip and acoustic velocity is the collective dynamics of the real contact area in contacts that are distributed in the bulk of the rock. Dynamics of the real contact area upon sliding has been studied in experiments with block contacts[54,60–62]. The real contact area decreases during sliding as new parts of the material are brought into contact. At rest, the real contact area increases as $\log(t)$ as described above for compressed contacts[60].

The corresponding $\log(t)$ increase of contact strength can be traced down to the atomic scale using atomic force microscopy[63]. The collective dynamics of a large ensemble of such discrete contacts was shown to govern the strength of a larger interface[61]. This is encompassed within empirical rate-and-state friction laws[64,65], which explicitly contain a $\log(t)$ strength increase if contact age is used as state parameter. Based on the large number of grain-to-grain contacts in the sandstone sample used in our experiments, we can expect that there are contacts which slide even at the small strains applied here.

Past testing on sandstones at even smaller strain levels (down to $10^{-8}$) has hinted at the fact that the nonclassical effect might only appear above a certain threshold, below which the sample would behave in a classically nonlinear elastic way[66]. This could be interpreted as further evidence that sliding along grain contacts, which only occurs once the frictional strength of the interface is overcome, can be held responsible for the nonclassical effect. Decreases of the acoustic wavespeed due to changes in the real contact area will thus occur whenever the sample is loaded or unloaded, and it will be followed by an increase with the characteristic $\log(t)$ dependence. Thus, the phenomenon of a real contact area decrease upon sliding along oblique contacts (Fig. 3) explains the instantaneous drop in velocity during both loading and unloading—as well as during an oscillating load—while the following re-increase along those same contacts during contact aging causes the subsequent recovery. Moreover, it also explains the vanishing anisotropy of the corresponding velocity signal.

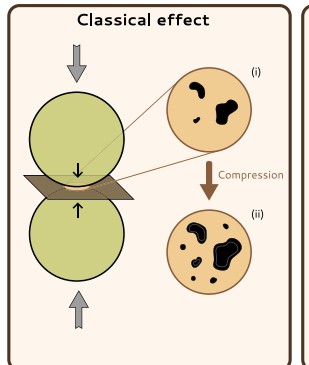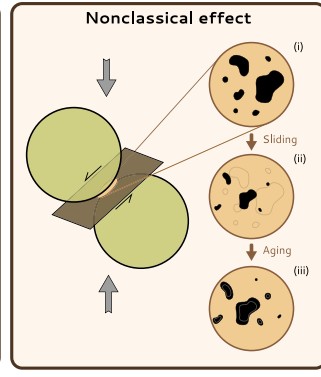

**Fig. 3 | Visual representation of the classical and nonclassical effects.** Schematic representation of grains with their nominal contact areas (yellow circles) and the corresponding contact planes (brown rectangles). The right-hand side of each panel is a magnified version of the nominal contact area, with the real contact area drawn in black. For the classical effect, the real contact area (black) increases between (i) and (ii) when uniaxial compression (gray arrows) is applied. Removing the load acts in the opposite direction. For the nonclassical effect, sliding along the inclined contact plane will both damage and displace contact points between (i) and (ii). During the subsequent recovery phase, when sliding has stopped, contact aging increases the contact area again. When removing the uniaxial compression, the same process occurs again since the direction of sliding is irrelevant for the damage, and the process from (i) to (iii) is repeated.

Let us take the two-dimensional case as an example to examine the anisotropy. The contacts sliding first have low friction and will roughly form a +45° angle with the loading axis. Similar to the classical effect, the change in real contact area affects the velocity as $\cos^2(\gamma)$, where $\gamma$ is the incidence angle of the ray. By the same token, the set of contacts at a -45° angle with the loading axis has a $\sin^2(\gamma)$ dependence. The overall angle dependency is $\cos^2(\gamma) + \sin^2(\gamma) = 1$, leaving us with an overall constant and isotropic nonclassical effect (ii).

We stress that our results indicate that the inclined sliding contacts themselves cause the nonclassical effect, and that the recovery does not rely on continued sliding of the contacts since contact aging takes place during hold. This means that wing cracks or similar opening and closing features at the edges of the sliding contacts cannot be responsible for the nonclassical damage and healing[40], as their opening and closing results in the same anisotropy and sign as the classical effect, even if the dynamics is mediated by friction of the sliding contact.

Our experiments show that the relative acoustic velocity variations have two main contributors: the classical component, which is correlated to the applied stress, and the nonclassical component, which is always a decrease during load changes irrespective of their sign followed by the SD $\log(t)$ recovery. By using direct wave velocity measurements with positive and negative stress changes at different angles with the loading axis, we introduce anisotropy as an explicit observable to study SD. We observe a decisive difference in anisotropy between the classical and nonclassical nonlinear elastic effects, manifesting differences in their origin.

Based on time dependency, symmetry with respect to the strain change, and, most importantly, on anisotropy, we propose that the nonclassical effect is due to frictional processes at oblique sliding contacts while the classical effect is due to compression of contacts oriented perpendicular to the loading axis.

This conclusion has two consequences. First, it provides a natural explanation for the characteristic and omnipresent shape of SD, namely an instantaneous decrease followed by a transient $\log(t)$ increase. The damage is directly imposed by the perturbation that displaces the opposite sides of contacts, known as contact

rejuvenation in the physics of friction[67]. The transient increase−or recovery−of the elastic moduli in the healing phase is related to the transient increase in frictional strength during hold time, known as contact aging in friction experiments[65]. Even though the precise mechanism is not clear (increase of contact area or contact quality, creation of chemical bonds, indentation creep), recognizing the common origin of SD and contact aging might ease future investigation. According to this interpretation, the nonclassical effect represents the earliest phase in the development of failure[1,68], when the damage is still localized in asperities distributed throughout the sample where it can recover on an observable time scale.

## Methods

### Material and sample preparation

We perform our experiments on a sample of Bentheim sandstone, retrieved near Bentheim, Germany. It is mostly composed of quartz (96.5%), followed by feldspar (2%), and kaolinite (1.5%). Grain size ranges from 50 μm to 550 μm and porosity is ~23.3%[47]. The reasons for our choice of material are twofold: first, it is a porous sandstone with grains that are homogeneous both in size distribution and in chemical composition. Second, it is very weakly anisotropic[42,69] by rock standards, with no discernible layering and no obvious grain shape anisotropy. This allows us to limit the complexity in our sample to its hard grains and soft bonds structure, which has been shown to be a key factor when investigating these nonclassical elastic effects[22].

The sample is inserted in a tightly fitting, impermeable neoprene jacket to insulate it from the confining medium (oil) in the triaxial cell. We cut holes in the jacket to directly glue brass casings (to hold P-wave piezoelectric transducers) onto the surface of the sample, then seal the gap, using two-component epoxy for both. The sample is then left to dry in a 50 °C oven for 12 h before carrying out the experiment.

### Mechanical testing

We carry out our experiment in a triaxial servo-hydraulic 4600 kN loading frame from Materials Test Systems, capable of applying both perfect hydrostatic pressure and vertical deviatoric stress on a sample[47]. The vertical perturbation is displacement-controlled using a Linear Variable Differential Transformer (LVDT). The apparatus has no temperature control. Compressive stress and strain are positive.

Initially, we set the confining pressure to 2 MPa and let the sample relax for 2 days. Then, maintaining the pressure throughout, we apply two uniaxial 5 μm ($\varepsilon = 5 \times 10^{-5}$) compression-decompression steps of varying durations, as described in Fig. 4. The LVDT is displacement-controlled throughout the experiment and held around the command level at all times. We refer to the periods where uniaxial strain is applied as the *compressed* state (colored background in Figs. 4 and 1), and those where only hydrostatic stress is being applied as the *relaxed* state.

### Active acoustics

Acoustic measurements are performed using a DAXBox transient recording system, to which we connect 14 P-wave piezoelectric transducers with a 1 MHz resonant frequency, each used as both ultrasonic transmitters and receivers. Each transducer sends out a 3 μs-long, 100 V rectangular electrical pulse every $T_\eta = 8$ s, and every other transducer picks up the propagated signal. The resulting waveforms are ~350 μs-long ultrasonic signals. Our acoustic system is designed to distinguish between active ultrasonics events and acoustic emissions[70], but we did not detect any of the latter, as the applied stresses are orders of magnitude below the elastic limit of Bentheim sandstone[43]. All traces are filtered using a third-order Butterworth band-pass filter between 700 kHz and 2 MHz. Due to the lack of synchronization between the transient recording system and the pulse generator, the source times are realigned during the pre-processing stage using the record of the pulsing sensor itself (Supplementary Discussion 1) For every source-receiver combination, we look at either the direct P-wave arrival, or the longest path containing exactly one reflection, based on the source-receiver geometry (see Supplementary Fig. 8). This allows us to avoid cases where surface and direct wave paths are of a similar length, and therefore overlap between their respective arrivals. Each individual path forms an angle $\theta$ with the loading axis. Purely longitudinal measurements could not be performed.

### Computing relative velocity changes

To compute the relative P-wave velocity change for sensor combination number $j$, we pick the first recorded trace $s_0^j(\xi) - \xi$ being the lapse time. We then compute a range of stretched versions $s_0^j(\eta; \xi) = s_0((1+\eta)\xi)$ of this reference trace for different values of the stretching coefficient $\eta$. We use cubic spline interpolation to obtain sub-sample precision. Then, for the $i$-th recorded trace $s_i^j(\xi)$, we crop a 10 μs-long time window centered around the estimated P-wave arrival, and cross-correlate (represented by $\cdot$ in Equation (2)) the cropped $\hat{s}_i^j(\xi)$ with the uncropped $s_0^j(\eta; \xi)$. Using this method, the optimal stretching coefficient $\eta_i^j$ is exactly equal to the relative velocity change $dv/v$ in the selected time window:

$$\eta_i^j = \arg\max_\eta \hat{s}_i^j(\xi) \cdot s_0^j(\eta; \xi) \tag{2}$$

Corrections to the measurements of $\eta$ for effects of deformation and stress control are provided in the Supplementary Discussion 1. Figure 1 shows three examples of observed velocity changes for waves propagating through the sample for different wave paths. These illustrate systematic differences between $\eta$ depending on $\theta$, which we quantify in the next sections.

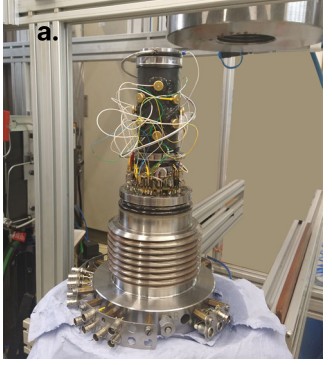

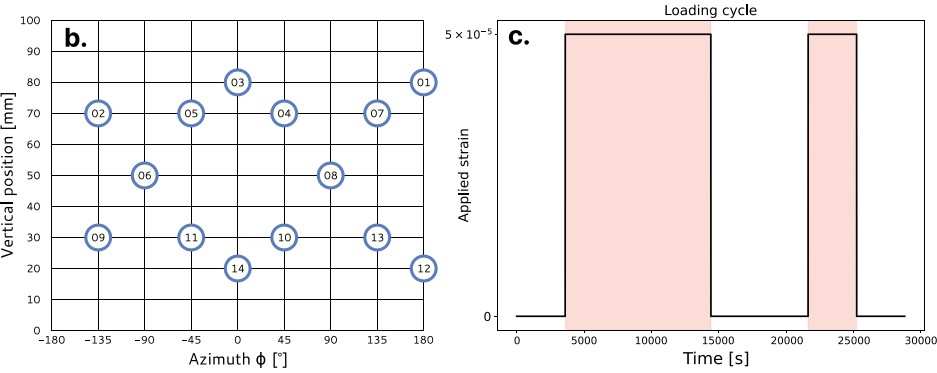

**Fig. 4 | Overview of the experimental setup. a** Photograph of the sample before its insertion in the cell. **b** A flat projection of the transducer locations. (blue circles) on our cylindrical sample **c**. Graph describing the loading cycle. The colored background indicates when uniaxial strain is being applied to the sample.

## Decomposition of classical and nonclassical elastic effects

We aim to quantify the amplitude of the instantaneous classical and transient nonclassical nonlinear effects as a function of $\theta$. To this end, we develop a model for $\eta$ which separately incorporates both a classical and a nonclassical component. For a given observation, this allows us to quantify the amplitude of the two effects with a single scalar value for each. We use the following decomposition of the relative velocity change $\eta$, which assumes that the different constituents are linearly superimposed:

$$\eta(t;\theta) = \eta_L(t) + \eta_C(t;\theta) + \eta_{NC}(t;\theta) = \alpha L(t) + \beta(\theta)C(t) + \delta(\theta)R(t) \quad (3)$$

where $t$ denotes the wall clock time. The linear component $L(t)$ describes a consistent linear increase in velocity across sensor combinations that is solely a function of experimental time. It represents a drift, common to all measurements, and which is most likely introduced by temperature changes that affect the experimental setup, even though the experiment was performed at night to minimize this effect. Its determination is detailed in Supplementary Discussion 2. We then have the classical component $C(t)$, which is nonzero and constant only when deviatoric stress is applied to the sample, and otherwise zero. Finally, the nonclassical component $R(t)$ is made from a series of the typical slow-dynamics softening-recovery sequences that start whenever an elastic perturbation is induced by changes in the axial loading. The three constituents of the model are illustrated in Fig. 1. The method for computing the uncertainties associated with each parameter are detailed in Supplementary Discussion 3.

The key building block for modeling the transient part of our $\eta$ measurements is the relaxation function $R^\star(t)$, defined in Equation (4) as

$$R^\star(t) = \int_{\tau_{min}}^{\tau_{max}} \frac{1}{\tau} e^{-\frac{t}{\tau}} d\tau \text{ for } t > 0 \quad (4)$$

This function[71] is designed to describe a system whose total observed relaxation results from the superimposition of a distribution of relaxing phenomena after a perturbation at $t = 0$, each with their own relaxation time $\tau$. Such a function is described by two parameters, $\tau_{min}$ and $\tau_{max}$, which define the range of relaxation times present in the system. While it is logarithmic for intermediate times between $\tau_{min}$ and $\tau_{max}$, the function $R^\star$ converges for both $t = 0$ and $t = +\infty$. It is also easily parameterizable: the choice of $\tau_{min}$ is dictated by the sampling period $T_\eta$ of the $\eta$ measurements since it has been shown that the $\log(t)$ recovery starts significantly sooner after a perturbation than the few seconds sampling interval $T_\eta = 8$ s of our observations. We therefore choose $\tau_{min} = 0.1$ s, knowing that there is no sensitivity to time scales significantly shorter than the sampling interval.

To estimate $\tau_{max}$, we used an iterative grid search over several orders of magnitude of $\tau_{max}$ to test whether there is any sensitivity to the longest relaxation processes. We find that the data can be better fit by increasing $\tau_{max}$ up to about $10^4$s, beyond which no discernible improvement is possible. This indicates that the longest relaxation timescales are longer than the hold times in the experiment. Since there is no sensitivity in our measurements to longer relaxation times, we set $\tau_{max} = 1.5 \times 10^4$s. Using this relaxation function as a base, we build the transient component for the modeling:

$$R(t) = R^\star(t - t_i) \text{ for } t_i < t < t_{i+1} \quad (5)$$

where the $t_i$ denote the times at which we change the stress either by loading or unloading the sample.

Figure 1 shows three different examples of fits that we achieve between the model and the data for different values of $\theta$. While, by construction, the linear component is the same for all observations, the contributions of the classical and nonclassical components differ between observations, and are quantified by the parameters $\beta(\theta)$ and

$\delta(\theta)$, respectively. For the sake of argument, we study the possibility of fitting the stress states individually using a 10-component model in the Supplementary Discussion 4, and argue that there is no significant advantage in doing so.

## Data availability

The relative velocity change data and the inverted model coefficients for each sensor combination along with the mechanical data generated in this study are attached as an electronic supplement in the Supplementary Software file. Source data are provided with this paper.

## Code availability

The functions used to display and compute the relative velocity changes as well as invert the model parameters from the relative velocity change measurements are available as an electronic supplement in the Supplementary Software file, along with a Jupyter notebook for visualization purposes.

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

## Acknowledgements

The authors thank Stefan Gehrmann, Lei Wang and Michael Naumann for their assistance during experiments; Peter Makus and Grzegorz Kwiatek for their contribution to the code; Céline Hadziioannou, Marco Dominguez Bureos and Zihua Niu for helpful discussions. This project has received funding from the European Union's Horizon 2020 research and innovation program under the Marie Skłodowska-Curie grant agreement No.955515, the SPIN (*Seismological Parameters and INstrumentation*) Innovative Training Network. Open Access funding enabled and organized by the DEAL Consortium.

## Author contributions

M.A., C.S., and A.B. conceived the methodology. M.A. and A.B. carried out the experimental work. M.A., C.S. and A.C. contributed to the modeling of the data. M.A., C.S., A.B. and A.C. contributed to the interpretation of the results. G.D. and M.B. supervised the work. M.A. and C.S. wrote the first draft. All co-authors revised the final draft of the manuscript.

## Competing interests

The authors declare no competing interests.
