## [Transparent Peer Review file · Nature Communications]

Anisotropy reveals contact sliding and aging as a cause of post-seismic velocity changes

Corresponding Author: Mr Manuel Asnar

Version 0:

Reviewer comments:

Reviewer #1

(Remarks to the Author)

This work describes a theory for why two phenomena observed in nonlinear elasticity behave differently. It is important for the nonlinear elasticity field to understand this and peripherally important to seismology in more generality.

This is a really interesting experiment, and the results are clearly presented and well illustrated. The writing is good, and the conclusions correspond with the results. The work is well justified, and the supplemental information goes through most of the obvious questions one might have about the data and results. That said, I think that there is some missing literature review that if properly taken into account weakens the novelty and impact of the results. I will start with that and then give my few scientific questions.

For the literature, from 2016, you have:

Rivière, J., L. Pimienta, M. Scuderi, T. Candela, P. Shokouhi, J. Fortin, A. Schubnel, C. Marone, and P. A. Johnson (2016), Frequency, pressure, and strain dependence of nonlinear elasticity in Berea Sandstone, *Geophys. Res. Lett.*, 43, 3226–3236, doi:10.1002/2016GL068061.

In that paper, they explicitly state that they see two separate mechanisms. They do not speculate (much) as to what these mechanisms are and perhaps your 'tilted-sliding contacts' would explain the differences that they see, but this deserves for sure a reference as well as a significant discussion of how your results fit into theirs.

You do not mention this paper:

Bittner & Popovics *Appl. Phys. Lett.* 114, 021901 (2019)

in which they directly image fluid moving out of a pore during nonlinear excitation. How does that fit within your model? At face value it seems to contradict your results in that they clearly see slow dynamics showing a large effect on fluids in the pore space.

This paper was I think the first to really discuss anisotropy in nonlinear experiments:

TenCate, James A., et al. "The effect of crack orientation on the nonlinear interaction of a P wave with an S wave." *Geophysical Research Letters* 43.12 (2016): 6146-6152.

Overall my conclusion is that the work could use a bit more context.

My only real scientific concern is the spread in your data. I know how sensitive these measurements are and how small the signals are that you are trying to extract, so the spread in the data is understandable. But it is large, and that does make it consistent with a lot of different models. What does a straight line fit look like instead of a cosine? Do your data rule that model out? It does look like there is a trend in the classical effect and perhaps not in the non-classical, but I'm not convinced that your model is correct, or the only one that can fit your data.

Now for some minor suggestions:

In the abstract, you state: "while the loading effect shows the expected anisotropy originating from the opening and closing of cracks." It is not clear to me what the 'loading effect' is, what the expected anisotropy in this would be, and even after reading the paper I am not sure in what way you have shown this.

You repeat 'concentrate' in line 3.

Line 22 I think you mean hardening not softening.

Line 23 you say: "which is linear with respect to the logarithm of time elapsed" Just say it has $\log(t)$ dependence and then define t .

I find it hard to compare the curves for the different angles in Figure 1. I see that they would be on top of one another if you plot the data without the offsets, but perhaps you could plot all of the models on top of one another to let the reader better see the differences between the various curves.

In figure 1, the caption should also mention the second panel.

In the supplemental information, you derive a simple model that has similar dependencies to your model, but you say that you cannot fit it to your data because you do not have values for the third-order elastic constants. Could you not fit the data to that model to derive those? Or at least plot the model for a guess at those parameters? If I recall this paper:

Renaud G, Talmant M and Marrelec G 2016 Microstrain-level measurement of third-order elastic constants applying dynamic acousto-elastic testing J. Appl. Phys. 120 135102

has some reasonable values that one could use.

It is not clear to me what path the reflected waves follow. Are they going straight across and back? Perhaps a small diagram would be helpful for this.

(Remarks on code availability)

Reviewer #2

(Remarks to the Author)

The manuscript was difficult to read and the significance of the results, therefore, unclear. I am afraid any novelty of this work will not reach people outside of the author's scope. I have noted several issues in the initial sections of the manuscript below.

Line 3 - revise "concentrating and concentrates"

Line 7 - a leading sentence of a paragraph shouldn't refer to "it". What is "it"?

Line 9 - "this process", similar to previous comment.. the language needs to be more exact

Line 10 - it isn't clear whether the authors are listing three separate mechanical properties or two to three different ways to describe the same property.

Line 12 - reference to "originally coined by the acoustics community"?

Line 13 - "this term" Revise to be more explicit. Slow dynamics, originating from the acoustics community, describes a transient process...

Line 14 - does perturbation refer to a local static or dynamic strain? The perturbation is related to the heterogeneity or is a homogeneous/far-field strain also included as a type of perturbation?

Lines 20, 21 - perturbation is not well defined and so what does "sign of perturbation" mean? Please think of a general reader. If one tries to follow from the previous sentences, then perturbation refers to strain. A change in sign of strain refers to a transition through the 0 strain state (tension into compression). I don't think the authors want the readers to think in this direction. Rather, the change in "sign" is related to the compression/decompression stages or the heating/dissipation stages, etc.

>> A figure clearly illustrating compression/decompression and corresponding strain vs time plot would be helpful to the reader

Lines 20-22 - it sounds like the sentence reads as if softening never occurs during slow dynamics. Are the authors trying to

say something along the lines of... "The slow dynamics response (which includes softening and recovery) occurs regardless of the sign of the perturbation. This means that loading/unloading and heating/cooling all trigger the same response pattern - they all cause transient softening followed by logarithmic recovery."?

Line 26 - "comparatively" compared to what?

Lines 29-33 - These lines attempt to define classical and nonclassical. The authors make it appear that "classical" is all encompassed by finite strain theory. This appears to be restrictive. Where would micromechanical theories fit? A better description of "classical" is needed.

Lines 51-58 - What is the physical difference between a "pump wave" and a uniaxial compression/decompression test within the context of the paper? Is it the time-dependence ($e^{i\omega t}$)?

Line 59 - a previous sentence made claim to SD not being present at high pressures. How can this work observe slow dynamics at 80 MPa if that is the case? Or are you saying that study was not a "true" study of SD since it is not in the right pressure range? Please clarify.

Line 70 - Related to my comment in Lines 51-58. If a pump wave can't be used to probe SD then how can the dynamic modulus of a P-wave?

Line 96 - the experiments test only a single flavor of anisotropy, namely, transversely isotropic behavior. Claims about independence of "anisotropy" are too general.

(Remarks on code availability)

Reviewer #3

(Remarks to the Author)

I found the introduction to be particularly well written in describing our state of knowledge in the context of the main question addressed in the experiments and analysis.

On line 3, "structures, concentrating and concentrates". This needs rewording.

On line 26, you write "Remarkably, SD is most apparent for comparatively low pressures (0.1MPa to 10MPa) and applied strains (10^{-4} to 10^{-7}) before vanishing at higher amplitudes." Do you mean higher amplitudes of pressure, or strains here? I assume you mean pressures.

Additionally on line 208, you write "Moreover, testing on sandstones at even smaller strain levels (10^{-6} to 10^{-8}) clearly shows that the nonclassical effect only appears above a certain threshold, below which the sample only behaves in a classically nonlinear elastic way" I would not put such precise boundaries on this behavior because there are other factors, such as frequency of the applied strains. The cited experiments do not fully explore the parameter space. These values are frequently cited but only apply to the experimental protocols that are used.

It looks from the experimental setup that 40 degrees may be the minimum angle that you could measure. It would be interesting to be able to go to 0 degrees. Based on your anecdote at line 222, the contacts at ± 45 degrees from the applied axial strain will slide first but will not produce any anisotropy. I agree that the classical and nonclassical behaviors are different in Fig. 2 but am just wondering if the apparent small increase in effect with decreasing angle continues to lower angles.

Could you reproduce Figure 1 (top) with the modeled $L(t)$ removed, perhaps in the supplemental? That would help to visualize just $R(t)$ and $C(t)$. Even better, could you show the data three times, each time showing only one of the three effects ($L(t), R(t), T(t)$) by subtracting the modeled values of the other two? This is similar to what is shown in the bottom half of the figure, except with the data points.

Also in Figure 1, the modeled $R(t)$ overshoots the observed nonclassical effect at the transitions from one stress state to another. Is this a sampling issue or something else?

Are the values in Figure 1 (bottom) normalized individually, or are their relative values accurate to your model? It doesn't look like that have the same DC offset. It would be nice to see them to scale and with the same DC offset.

Overall, I find your conclusions plausible and supported by the data. It would be interesting to see this tested on different and more complex materials. As the authors state, the origin of slow dynamics is an important topic in nonlinear elasticity. So, I think this manuscript will be of great interest to the community.

(Remarks on code availability)

I didn't run the code, but read through it. The code is well organized and documented.

Version 1:

Reviewer comments:

Reviewer #1

(Remarks to the Author)

Overall the authors have done an excellent job at addressing the vast majority of my concerns. I still think that the paper audience may be a bit narrower than what Nature aims at, but that is for the editors to decide. In terms of final suggestions:

- 1) I think that 'possible causes' rather than 'as a cause of' would be more accurate for the title
- 2) Lines 103-106 I suggest using an equation, or re-phrasing so that you define $C(t)$ and $R(t)$ before discussing how you add them together etc.
- 3) The sentence from lines 239-241 beginning "This behaviour is well-described" is awkward and not completely grammatical.

Best of luck!

(Remarks on code availability)

Reviewer #2

(Remarks to the Author)

Thanks for addressing my comments and incorporating sufficient revisions.

(Remarks on code availability)

Reviewer #3

(Remarks to the Author)

According to your interpretation, that nonclassical effects are due to contact sliding and aging. I'm wondering if there would be an anisotropy in the nonclassical elasticity when there is an anisotropic stress field applied. In your experiment, there is an applied uniaxial load during two intervals interspersed by periods of no load. During the period when the uniaxial load is applied, sliding surfaces that have a normal with a smaller angle with the load will have a higher normal stress and therefore age faster than sliding surfaces with higher angles with the load. There should be a diminishing amount of slip on surfaces as the normal approaches 0 and 90 degrees. P-waves traveling at a lower angle would be slightly faster than those traveling at a higher angle due to different rates of aging. This would not be the case when the load is removed because the stress field would no longer be anisotropic. One way to test this is to see if the small amount of anisotropy that you measured for the nonclassical component all comes from measurements taken when the stress field is anisotropic. You could reproduce Figure 2 using only points when the load is applied and again using only points when load is not applied and see if they are different.

You state that SD vanishes at higher pressures (line 29), cited from a previous work. I guess I would say that it still exists but aging or relaxation happens so fast that we don't observe it. I'm not suggesting you need to change this sentence because you are simply reporting a previous work, but I think it is saying that aging rate increases with confining stress.

(Remarks on code availability)

As I wrote in my first review, I read through, but did not run the code. It was easy to understand what they were doing.

Authors' Response to Reviews of

Anisotropy reveals contact sliding and aging as a cause of post-seismic velocity changes

Manuel Asnar, Christoph Sens-Schönfelder, Audrey Bonnelye, Andrew Curtis, Georg Dresen, Marco Bohnhoff

RC: Reviewers' Comment, AR: Authors' Response, □ Manuscript Text

1. Reviewer 1

1.1. Major comments

RC 1: *This work describes a theory for why two phenomena observed in nonlinear elasticity behave differently. It is important for the nonlinear elasticity field to understand this and peripherally important to seismology in more generality.*

AR: We appreciate the Reviewer's acknowledgment of the relevance of this work.

RC 2: *This is a really interesting experiment, and the results are clearly presented and well illustrated. The writing is good, and the conclusions correspond with the results. The work is well justified, and the supplemental information goes through most of the obvious questions one might have about the data and results. That said, I think that there is some missing literature review that if properly taken into account weakens the novelty and impact of the results. I will start with that and then give my few scientific questions.*

For the literature, from 2016, you have Rivière, J., L. Pimienta, M. Scuderi, T. Candela, P. Shokouhi, J. Fortin, A. Schubnel, C. Marone, and P. A. Johnson (2016), Frequency, pressure, and strain dependence of nonlinear elasticity in Berea Sandstone, Geophys. Res. Lett., 43, 3226–3236, doi:10.1002/2016GL068061. In that paper, they explicitly state that they see two separate mechanisms. They do not speculate (much) as to what these mechanisms are and perhaps your 'tilted-sliding contacts' would explain the differences that they see, but this deserves for sure a reference as well as a significant discussion of how your results fit into theirs.

AR: We thank the Reviewer for reminding us of the discussion of the distinct effects in the paper they mention. In said paper, they do indeed state explicitly that they see two distinct mechanisms: one that is frequency-independent (which we may attribute to the classical effect, which is time-independent), and one that is frequency-dependent. When discussing which physical mechanisms might be the ones causing these frequency-dependent components, however, they mention the opening and closing of cracks. This is incompatible with the distinct pattern of anisotropy that we report. As you point out, we could reasonably expect sliding along tilted contacts to explain their observations.

We have added a reference to this paper along with several others to discuss previous suspicions about the existence of two distinct mechanisms. However, our contribution is to provide evidence for the fact that these two mechanisms actually occur at different structures which helps to identify the processes that actually happen.

Moreover, A number of past studies have already hinted at there being two fundamentally distinct types of nonlinear elastic behavior [21,26,27,39], which often appear together in experiments. While

it is commonly agreed that the opening and closing of cracks is responsible for a significant portion of the nonlinearity, it is unclear how it can generate two distinct behaviors by itself. As a result, many experimental studies are left with part of their observations being inconsistent with this explanation: for example, higher-frequency fluctuations in an otherwise low-frequency nonlinear response [39], or a single pump frequency-independent harmonic component in the nonlinear response [21]. Moreover, the $\log(t)$ recovery of the nonclassical component can only be explained by an additional time-dependent process, which exclusively affects the structures responsible for the nonclassical effect, as opposed to the classical effects which are time-independent [21].

RC 3: *You do not mention this paper: Bittner & Popovics Appl. Phys. Lett. 114, 021901 (2019), in which they directly image fluid moving out of a pore during nonlinear excitation. How does that fit within your model? At face value it seems to contradict your results in that they clearly see slow dynamics showing a large effect on fluids in the pore space.*

AR: For brevity, we intended to avoid discussions of the influence of moisture and did therefore not discuss this very nice work. We have now added a reference to it in the manuscript. In fact, we think that these observations are very compatible with the concept that we put forward. In the conclusion, the authors themselves admit that they can only speak of a correlation between the alleged fluid migration and the recovery process. In their follow-up 2021 paper (which we do cite in our manuscript), they design a model where fluid vaporization during conditioning and subsequent diffusion and condensation at grain contacts during the relaxation phase are held responsible for the slow dynamics recovery. The water is likely not squeezed out of initially filled pores due to compaction, but is rather evaporated from grain surfaces due to motion of contacts. Just assuming that the motion can be due to shear along tilted contacts would reconcile our theory with Bittner's observations.

Models have been proposed attributing slow dynamics to the adsorption of fluid at the contacts between grains [30][28,29], the destruction and recreation of bonds in contact asperities [30], or more generally to internal damage processes [31,32].

RC 4: *This paper was I think the first to really discuss anisotropy in nonlinear experiments: TenCate, James A., et al. "The effect of crack orientation on the nonlinear interaction of a P wave with an S wave." Geophysical Research Letters 43.12 (2016): 6146-6152.*

AR: This is also a relevant reference. They are looking at the effect of crack orientation on the overall nonlinear elastic behavior of very anisotropic sandstone, and rightly conclude that the nonlinearity is higher when particle motion is perpendicular to the bedding planes - and therefore to the crack orientations. However, they do not distinguish between the classical and nonclassical nonlinearities, which makes it hard to draw conclusions about their respective physical origins. Additionally, while in their measurements the envelopes are of different amplitudes, the so-called high-frequency ripples - for which they themselves say that it is unlikely that the opening and closing and cracks would be responsible - are of comparable amplitudes for both orientations, which could be consistent with our claim that the nonclassical nonlinearity is isotropic. They also do not look at the recovery period explicitly. We have included a mention to them in our introduction and briefly discuss their results.

The effect of crack orientations in anisotropic sandstones on nonlinear elastic effects has also been investigated [39]. However, no distinction between classical and nonclassical effects was made in that study, and only the conditioning phase was investigated. Therefore, despite observing a significant dependence of the amplitude of nonlinear effects on crack orientation in the sample, one cannot draw conclusions about the transient effect alone.

RC 5: *Overall my conclusion is that the work could use a bit more context.*

AR: Starting from the references that the Reviewer brought to our attention, we have added further context and discussion as described in this point-by-point response.

RC 6: *My only real scientific concern is the spread in your data. I know how sensitive these measurements are and how small the signals are that you are trying to extract, so the spread in the data is understandable. But it is large, and that does make it consistent with a lot of different models. What does a straight line fit look like instead of a cosine? Do your data rule that model out? It does look like there is a trend in the classical effect and perhaps not in the non-classical, but I'm not convinced that your model is correct, or the only one that can fit your data.*

AR: We understand your concerns about the spread of our data. As you say, these measurements are highly sensitive, and the $u \cos^2(\theta) + v$ model that we picked might not be the only one that fits it. We did in fact initially attempt to fit a simple straight line to our data (which we replicated and included in the Supplementary Material), which performs just as well as the cosine in showing the differences in anisotropy between the two effects. However, we investigate an angular dependency here for which, due to symmetry, we can assume some kind of periodicity. This cannot be accomplished with a linear fit. Without insisting on the cosine law, we opted for an anisotropy law which had already been linked to induced anisotropy in rocks. The only important fact (which is also obtained with a linear fit) is that the classical effect is anisotropic (as commonly known) while the nonclassical effect is significantly and distinctly less anisotropic.

We have provided an additional discussion about this in the Supplementary Material (see Supplementary Discussion 4), as well as a reference to it in the main manuscript.

1.2. Minor comments

RC 7: *Now for some minor suggestions. In the abstract, you state: "while the loading effect shows the expected anisotropy originating from the opening and closing of cracks." It is not clear to me what the 'loading effect' is, what the expected anisotropy in this would be, and even after reading the paper I am not sure in what way you have shown this.*

AR: Thank you for pointing that out. We replaced the word "loading" by "acoustoelastic", as this was actually what we meant to say. As for what its expected anisotropy should be, we investigate it in detail in the Supplementary Material, specifically in the Supplementary Discussion 4. The classical component in our model is essentially the acoustoelastic part of the sample's response, and we show that it exhibits anisotropy which is consistent with what previous studies and finite strain theory would predict - namely the $\cos^2(\theta)$ dependence.

RC 8: *You repeat 'concentrate' in line 3.*

AR: We rephrased it.

RC 9: *Line 22 I think you mean hardening not softening.*

AR: We rephrased the sentence for clarity.

Remarkably ~~so~~, the slow dynamics response to a ~~perturbation is independent from the sign of the perturbation~~mechanical or thermal perturbation is symmetry-breaking; static loading and unloading as well as heating and cooling ~~never cause transient~~always cause a sharp softening followed by transient hardening, and never a sharp hardening followed by transient softening. A hallmark of the slow dynamics recovery is the shape of its temporal evolution, which is linear with respect to the logarithm of time elapsed since the ~~perturbation [11,12] - which end of the perturbation [4,11]~~. This is why the phenomenon is commonly referred to as *slow*, as opposed to the *fast* dynamics of the perturbation.

RC 10: *Line 23 you say: "which is linear with respect to the logarithm of time elapsed" Just say it has log(t) dependence and then define t.*

AR: We understand that the sentence comes across as a little verbose. However, we are not convinced that phrasing it in a different way would necessarily make it clearer, and the precise definition of t in that context - that is, the precise time at which the relaxation "starts", is very much still an open research question. Since this particular point does not seem to have raised any issues among the other reviewers nor among our authors, we would therefore like to keep it phrased as is.

RC 11: *I find it hard to compare the curves for the different angles in Figure 1. I see that they would be on top of one another if you plot the data without the offsets, but perhaps you could plot all of the models on top of one another to let the reader better see the differences between the various curves.*

AR: Indeed, the data without the offsets overlaps and, although it makes for a fitting illustration of our paper's thesis, we found it to be detrimental to the clarity of the plot. We have added a version of Figure 1 without the offsets in the Supplementary Material.

RC 12: *In figure 1, the caption should also mention the second panel.*

AR: We added a description of the second panel to the caption.

Relative velocity change measurements and model components a) Relative velocity change measurements η for three different values of θ . ~~More examples are provided in Extended Data Figure 1.~~ Solid lines show the respective model fits. The ~~colored background corresponds to periods where uniaxial strain is applied on the sample.~~ The origin of each set of measurements ~~is has been~~ offset by 0.015 % for ~~easier visualization~~ clarity. b) The individual components for the linear inversion ~~are~~ as described in Equation 3. ~~The components are of unit amplitude, so that the corresponding coefficients α , β and δ can be compared. The mean of each component is subtracted prior to the linear inversion.~~

RC 13: *In the supplemental information, you derive a simple model that has similar dependencies to your model, but you say that you cannot fit it to your data because you do not have values for the third-order elastic constants. Could you not fit the data to that model to derive those? Or at least plot the model for a guess at those parameters? If I recall, Renaud G, Talmant M and Marrelec G 2016 Microstrain-level measurement of third-order elastic constants applying dynamic acousto-elastic testing J. Appl. Phys. 120 135102 has some reasonable values that one could use.*

AR: We would like to thank the Reviewer for bringing to our attention this paper with TOEC for Berea sandstone. We have added corresponding estimates for u and v to the Supplementary Material. We also fixed our formulae in the Supplementary Material to be consistent with a positive compressive stress convention.

RC 14: *It is not clear to me what path the reflected waves follow. Are they going straight across and back? Perhaps a small diagram would be helpful for this.*

AR: Yes, those are the paths we chose for the reflected waves. We have added some diagram examples to the Supplementary Material, and a mention to it in the Methods section of the main manuscript.

2. Reviewer 2

RC 15: *The manuscript was difficult to read and the significance of the results, therefore, unclear. I am afraid any novelty of this work will not reach people outside of the author's scope. I have noted several issues in the initial sections of the manuscript below.*

AR: We understand the Reviewer's concerns and thank them for their comments. Although we are confident in our results and methodology, we hope the language adjustments we have made will help get our conclusions through to a wider audience.

RC 16: *Line 3 - revise "concentrating and concentrates"*

AR: We rephrased it.

RC 17: *Line 7 - a leading sentence of a paragraph shouldn't refer to "it". What is "it"?*

AR: We rephrased for clarity. "it" refers to the aforementioned processes of stress transfer and concentration.

Depending on ~~the stage of this process~~, it how far along it is, this process of stress transfer and concentration can be observed through changes in stiffness, macroscopic deformation, or acoustic emissions.

RC 18: *Line 9 - "this process", similar to previous comment.. the language needs to be more exact*

AR: We rephrased it. See response to the above comment.

RC 19: *Line 10 - it isn't clear whether the authors are listing three separate mechanical properties or two to three different ways to describe the same property.*

AR: We are listing three mechanical properties which, although very closely linked, are generally referred to as distinct.

RC 20: *Line 12 - reference to "originally coined by the acoustics community"?*

AR: We added a reference to the 1996 TenCate paper, "Slow dynamics in the nonlinear elastic response of Berea sandstone", which to our knowledge is the first to name this phenomenon. See the response to the comments below.

RC 21: *Line 13 - "this term" Revise to be more explicit. Slow dynamics, originating from the acoustics community, describes a transient process...*

AR: We added "latter" for clarity while avoiding repetition.

RC 22: *Line 14 - does perturbation refer to a local static or dynamic strain? The perturbation is related to the heterogeneity or is a homogeneous/far-field strain also included as a type of perturbation?*

AR: By "perturbation" we mean any kind of external mechanical or thermal influence exerted on the sample - usually acting on it in the form of strain. This can take the form of dynamic oscillations, heating the sample via a laser, sending large-amplitude shear waves through it, applying incremental static loads to it ... We do not go into much further detail in our attempt to define "perturbation" at this stage, and rather rely on the later enumeration of past experimental studies of slow dynamics in order to impress on the reader what shapes such a perturbation might take. We did however slightly rephrase in order to specify what we meant by "perturbation".

This latter term, originally coined by the acoustics community [4], describes a transient process of stiffness recovery after a softening that is induced by almost any type of static or dynamic **strain**

mechanical or thermal perturbation.

RC 23: *Lines 20, 21 - perturbation is not well defined and so what does "sign of perturbation" mean? Please think of a general reader. If one tries to follow from the previous sentences, then perturbation refers to strain. A change in sign of strain refers to a transition through the 0 strain state (tension into compression). I don't think the authors want the readers to think in this direction. Rather, the change in "sign" is related to the compression/decompression stages or the heating/dissipation stages, etc.*

AR: We rephrased this in terms of symmetry-breaking rather than sign-independence. See response to Reviewer Comment 9.

RC 24: *A figure clearly illustrating compression/decompression and corresponding strain vs time plot would be helpful to the reader*

AR: In order to limit the number of figures in the main part of our paper, and since the loading procedure was relatively simple, we have opted to simply color the background in the relevant figures during the periods of uniaxial compression. An additional figure illustrating compression/decompression and corresponding strain vs time plot was also already provided in the Methods section of the manuscript (Figure 4). We added clarifications in the main text as well as in the caption of Figure 1.

RC 25: *Lines 20-22 - it sounds like the sentence reads as if softening never occurs during slow dynamics. Are the authors trying to say something along the lines of... "The slow dynamics response (which includes softening and recovery) occurs regardless of the sign of the perturbation. This means that loading/unloading and heating/cooling all trigger the same response pattern - they all cause transient softening followed by logarithmic recovery."?*

AR: We rephrased it. See response to Reviewer Comment 9.

RC 26: *Line 26 - "comparatively" compared to what?*

AR: We meant "comparatively" as in "fairly". We rephrased it.

RC 27: *Lines 29-33 - These lines attempt to define classical and nonclassical. The authors make it appear that "classical" is all encompassed by finite strain theory. This appears to be restrictive. Where would micromechanical theories fit? A better description of "classical" is needed.*

AR: We do in fact define "classical" as the part of nonlinear elastic behavior that can be covered by finite strain theory, or what is sometimes referred to as acoustoelasticity. Some micromechanical theories are mentioned further below, but most such theories that we are aware of tend to concern themselves with the nonclassical part of the effect, namely the hysteresis and time-dependency. It is arguably quite restrictive, but we do believe that our experiment, model and results along with a number of past studies all argue in favor of drawing the line there. We have added a reference to acoustoelasticity to help clarify what we mean by "classical".

Although SD is linked to nonlinearity in the elastic properties [11], the transient softening and recovery cannot be reproduced in the frame of classical nonlinear elasticity, sometimes referred to as acoustoelasticity or finite strain theory [24,25]. The presumed existence of two different types of nonlinear elastic behavior in rocks has been mentioned in past studies [21,26,27]. Therefore, in contrast to the instantaneous classical nonlinear effects, we refer to processes involving a time dependency – such as slow dynamics – as nonclassical effects.

RC 28: *Lines 51-58 - What is the physical difference between a "pump wave" and a uniaxial compression/decompression test within the context of the paper? Is it the time-dependence ($e^{i\omega t}$)?*

AR: Apologies for the lack of clarity here. The time-dependence is indeed the main factor here: what we referred to as the "pump" when discussing the paper actually corresponds to the period during which dynamic strain

oscillations are applied to the sample, sometimes referred to as "conditioning". In our paper, this would correspond to the time period during which the strain that is being applied to the sample is changed. This, in our case, happens over a few seconds during each stress change. We rephrased this to make it a bit clearer and get rid of the word "pump".

Additionally, this study investigated the sample during the ~~action of the pump wave~~ dynamic perturbation phase only, and without analyzing the relaxation phase that starts once the perturbation stops, hindering the ability to investigate the transient effect separately.

RC 29: *Line 59 - a previous sentence made claim to SD not being present at high pressures. How can this work observe slow dynamics at 80 MPa if that is the case? Or are you saying that study was not a "true" study of SD since it is not in the right pressure range? Please clarify.*

AR: We agree that it certainly does look like slow dynamics. But the statement that SD vanishes with increasing confining pressure is valid for a constant perturbing strain amplitude. The mentioned high-pressure experiments were performed at higher confining pressures, but also at larger differential stresses that might activate different deformation mechanisms, like creep. So indeed, we think that the experiments share important similarities with SD observations in acoustics and we include them to illustrate how many facets slow dynamics has as a near-universally occurring phenomenon in rocks and geomaterials. We specify this in the manuscript.

A lack of anisotropy of SD has been observed *en passant* in limestone [14] and in granite [40] at comparatively high pressures of 80 MPa, when acoustics experiments rarely go beyond 10 MPa. Friction and slip along grain boundaries and cracks was invoked to explain the transient effects [40] ~~but the applied strains~~. While these experiments were performed with confining pressure ranges about 10 times larger than the upper limit mentioned above, the applied strains also significantly exceed the usual ~~orders of magnitude strains~~ in acoustic experiments (10^{-4} to 10^{-7}) and field observations by two to five orders of magnitude.

RC 30: *Line 70 - Related to my comment in Lines 51-58. If a pump wave can't be used to probe SD then how can the dynamic modulus of a P-wave?*

AR: We apologize for the unclear phrasing. We did not actually mean that pump waves cannot be used to probe SD; rather, we meant that the healing part of SD is only dominant once the conditioning stops. It can then be probed by low-amplitude ultrasonic waves, which are used to infer changes in the dynamic modulus of the sample.

We removed any mention of a pump wave to avoid confusion (see response to Reviewer Comment 28), and clarified the role of the probe waves.

We investigate the anisotropy of both the instantaneous classical nonlinearity and the transient nonclassical nonlinearity using the relative seismic velocity changes η of compressional high-frequency, low amplitude compressional probe waves as a proxy for the dynamic modulus of the rock sample.

RC 31: *Line 96 - the experiments test only a single flavor of anisotropy, namely, transversely isotropic behavior. Claims about independence of "anisotropy" are too general.*

AR: It is true that we cannot guarantee with 100% certainty that the nonclassical effect is isotropic. However, the fundamental thesis of our paper is that the classical and nonclassical effects exhibit fundamentally irreconcilable types of anisotropy. From the literature, finite strain theory and our own model, we do know that the classical effect is transversely isotropic; and from our experiment we see that the nonclassical effect is most definitely not transversely isotropic, while it does seem to be isotropic. We cannot categorically rule

any kind of anisotropy, but we see a distinctly differing behavior and we do not see a reason to assume that our sample should exhibit some kind of exotic anisotropy when it comes to the nonclassical effect. We do look forward to seeing what future experiments on the topic might say on this specific aspect and believe that our work constitutes a significant first step in that direction.

3. Reviewer 3

3.1. Comments

RC 32: *I found the introduction to be particularly well written in describing our state of knowledge in the context of the main question addressed in the experiments and analysis.*

AR: We thank the Reviewer for their comment. We have polished the introduction even further, thanks to comments from Reviewer 1.

RC 33: *On line 3, “structures, concentrating and concentrates”. This needs rewording.*

AR: We rephrased it.

RC 34: *On line 26, you write “Remarkably, SD is most apparent for comparatively low pressures (0.1MPa to 10MPa) and applied strains (10⁻⁴ to 10⁻⁷) before vanishing at higher amplitudes.” Do you mean higher amplitudes of pressure, or strains here? I assume you mean pressures.*

AR: Yes, we clarified it in the manuscript.

RC 35: *Additionally on line 208, you write “Moreover, testing on sandstones at even smaller strain levels (10⁻⁶ to 10⁻⁸) clearly shows that the nonclassical effect only appears above a certain threshold, below which the sample only behaves in a classically nonlinear elastic way” I would not put such precise boundaries on this behavior because there are other factors, such as frequency of the applied strains. The cited experiments do not fully explore the parameter space. These values are frequently cited but only apply to the experimental protocols that are used.*

AR: We appreciate this remark. The values mentioned here were only describing the range of values over which the testing in the paper was actually carried out. They were not meant to be interpreted as hard boundaries on the nonclassical behavior. Rather, the main takeaway from this paragraph should be that some threshold might exist to begin with, which would be further evidence in favor of slip being a main driver of slow dynamics. We rephrased it in a way that we hope better reflects our reasoning.

Past testing on sandstones at even smaller strain levels (10⁻⁶ to 10⁻⁸) clearly shows down to 10⁻⁸ has hinted at the fact that the nonclassical effect only appears might only appear above a certain threshold, below which the sample only behaves would behave in a classically nonlinear elastic way [66]. This could be interpreted as further evidence that sliding along grain contacts, which only occurs once the frictional strength of the interface is overcome, can be held responsible for the nonclassical effect.

RC 36: *It looks from the experimental setup that 40 degrees may be the minimum angle that you could measure. It would be interesting to be able to go to 0 degrees. Based on your anecdote at line 222, the contacts at +/-45 degrees from the applied axial strain will slide first but will not produce any anisotropy. I agree that the classical and nonclassical behaviors are different in Fig. 2 but am just wondering if the apparent small increase in effect with decreasing angle continues to lower angles.*

AR: We were originally planning on measuring along the vertical axis ($\theta = 0^\circ$), but we could not access this data due to equipment malfunction. As far as the classical effect is concerned, according to our derivations in the Supplementary Discussion 5 and in line with previous observations, we would expect the increase in effect with decreasing angle to continue to lower angles, reaching a maximal value for 0° . For the nonclassical effect however, based on the suggestions we make in our manuscript, we would not expect it to increase significantly compared to higher angle values.

RC 37: *Could you reproduce Figure 1 (top) with the modeled L(t) removed, perhaps in the supplemental? That would help to visualize just R(t) and C(t). Even better, could you show the data three times, each time showing only one of the three effects (L(t),R(t),T(t)) by subtracting the modeled values of the other two?*

This is similar to what is shown in the bottom half of the figure, except with the data points.

AR: We have added the requested illustrations as a new figure in the Supplementary Material document. It is indeed a useful illustration.

RC 38: *Also in Figure 1, the modeled $R(t)$ overshoots the observed nonclassical effect at the transitions from one stress state to another. Is this a sampling issue or something else?*

AR: Yes, we do think it is a sampling issue. Our acoustic measurements are made every 8 seconds, and they are not necessarily synchronized with the stress state transitions. Therefore, although our $R(t)$ component is not strictly logarithmic, early-time fluctuations are still significantly faster than our 8-second sampling period.

RC 39: *Are the values in Figure 1 (bottom) normalized individually, or are their relative values accurate to your model? It doesn't look like that have the same DC offset. It would be nice to see them to scale and with the same DC offset.*

AR: The amplitudes and DC offsets of the linear components are displayed as we eventually use them when linearly inverting for our model parameters α , β and δ , with their mean removed. We updated the caption and y-axis label in Figure 1 to clarify this. See response to Reviewer Comment 12.

RC 40: *Overall, I find your conclusions plausible and supported by the data. It would be interesting to see this tested on different and more complex materials. As the authors state, the origin of slow dynamics is an important topic in nonlinear elasticity. So, I think this manuscript will be of great interest to the community.*

AR: Thank you for your interest. Indeed, we feel that a broader, more systematic study of the anisotropy of slow dynamics across different strains, pressures and materials could significantly contribute to our understanding of nonclassical effects in rocks, concrete and other complex materials.

3.2. Remarks on code availability

RC 41: *I didn't run the code, but read through it. The code is well organized and documented.*

AR: Thank you for your comments. We are glad that efforts to make the code readable and understandable were not made in vain!

Figure 1: **Relative velocity change measurements with different model components** Relative velocity change measurements η for three different values of θ . From top to bottom in each panel: 40° , 72° and 90° . Solid lines show the respective partial model fits, with the undesired components subtracted from the total model fit. Vertical offsets have been adjusted across panels for ease of reading.

Figure 2: **Relative velocity change measurements and model components** Relative velocity change measurements η for three different values of θ (see Figure 1 in the main manuscript). The origin of each scatter plot was aligned for easier comparison.

Figure 3: **Anisotropy of the classical and nonclassical effects** Results of the fitting of β and δ parameters to a law of the form $u(1 - \theta/\theta_0) + v$. Dots indicate measurements resulting from direct P-wave arrivals; stars are for reflected P-wave arrivals. Color of symbols shows uncertainty of individual measurements. The red curve is the average model with orange bands indicating the 95 % confidence interval. The histograms show the results of the 5000-iteration bootstrap sampling inversion of the u and v coefficients.

Figure 4: **Examples of reflected paths geometries** Some selected reflected ray paths that were included in our analysis for different sensor combinations, with the pulsing (green) and receiving (red) transducers and the reflection point (grey). Transducer numbers are as shown in Figure 4 of the main manuscript.

Authors' Response to Reviews of

Anisotropy reveals contact sliding and aging as a cause of post-seismic velocity changes

Manuel Asnar, Christoph Sens-Schönfelder, Audrey Bonnelye, Andrew Curtis, Georg Dresen, Marco Bohnhoff

RC: *Reviewers' Comment*, AR: Authors' Response, Manuscript Text

1. Reviewer 1

RC 1: *Overall the authors have done an excellent job at addressing the vast majority of my concerns. I still think that the paper audience may be a bit narrower than what Nature aims at, but that is for the editors to decide. In terms of final suggestions:*

1) I think that 'possible causes' rather than 'as a cause of' would be more accurate for the title

AR: We thank the reviewer for their encouraging words and reiterate our appreciation for their previous remarks, which helped us clarify our arguments and reasoning. Regarding the title suggestion, while it would also be a suitable option, we believe that our initial choice is ever so slightly better suited to the conclusions. This is because we think that the possibility of contact sliding as a mechanism for slow dynamics has been suggested before, and our analysis actually provides evidence for the fact that it is indeed a very likely contributor.

RC 2: *2) Lines 103-106 I suggest using an equation, or re-phrasing so that you define $C(t)$ and $R(t)$ before discussing how you add them together etc.*

AR: We added an additional mention to our model equation in the Results section and a short description in the caption of Figure 1.

RC 3: *3) The sentence from lines 239-241 beginning "This behaviour is well-described" is awkward and not completely grammatical.*

AR: We tweaked the corresponding sentence.

The corresponding $\log(t)$ increase of contact strength can be traced down to the atomic scale using atomic force microscopy [63]. The collective dynamics of a large ensemble of such discrete contacts was shown to govern the strength of a larger interface [61]. This ~~behavior is well-described by is encompassed within~~ empirical rate-and-state friction laws [64,65] ~~which explicitly contains, which explicitly contain~~ a $\log(t)$ strength increase if contact age is used as state parameter. Based on the large number of grain-to-grain contacts in the sandstone sample used in our experiments, we can expect that there are contacts which slide even at the small strains applied here.

RC 4: *Best of luck!*

AR: Thanks a lot for contributing to this paper!

2. Reviewer 2

RC 5: *Thanks for addressing my comments and incorporating sufficient revisions.*

AR: We thank the reviewer for their suggestions and are looking forward to sharing this improved version with the wider geophysics community.

3. Reviewer 3

RC 6: *According to your interpretation, that nonclassical effects are due to contact sliding and aging. I'm wondering if there would be an anisotropy in the nonclassical elasticity when there is an anisotropic stress field applied. In your experiment, there is an applied uniaxial load during two intervals interspersed by periods of no load. During the period when the uniaxial load is applied, sliding surfaces that have a normal with a smaller angle with the load will have a higher normal stress and therefore age faster than sliding surfaces with higher angles with the load. There should be a diminishing amount of slip on surfaces as the normal approaches 0 and 90 degrees. P-waves traveling at a lower angle would be slightly faster than those traveling at a higher angle due to different rates of aging. This would not be the case when the load is removed because the stress field would no longer be anisotropic. One way to test this is to see if the small amount of anisotropy that you measured for the nonclassical component all comes from measurements taken when the stress field is anisotropic. You could reproduce Figure 2 using only points when the load is applied and again using only points when load is not applied and see if they are different.*

AR: This is addressed to a certain extent in the Supplementary Discussion 4.2 and in the Supplementary Figure S7. There, we provide model fits where we have used separate components for each different stress state. The δ_1 and δ_3 components correspond to an anisotropic stress state, while the δ_2 and δ_4 correspond to the state without an applied uniaxial load. It is quite clear that they all exhibit very similar types and degrees of anisotropy regardless of the current stress state. As for your proposed explanation, it seems intuitive that additional normal load increases the healing rate, but it might also prolongate the process. So while we agree that there are reasons for imperfect isotropy of the nonclassical effect, we do not have clear theory for the effect of normal load changes on healing rates.

RC 7: *You state that SD vanishes at higher pressures (line 29), cited from a previous work. I guess I would say that it still exists but aging or relaxation happens so fast that we don't observe it. I'm not suggesting you need to change this sentence because you are simply reporting a previous work, but I think it is saying that aging rate increases with confining stress.*

AR: This is certainly an aspect that deserves a more in-depth study to determine whether the diminishing observability of slow dynamics with increasing pressure is due to a decreasing amplitude of the phenomenon with its rate staying the same, or if it is indeed because the healing rates have become so large that they become too fast to observe.

RC 8: *As I wrote in my first review, I read through, but did not run the code. It was easy to understand what they were doing.*

AR: Thank you again for your comments. We hope that our code will be of value to potential readers and did our best to make it accessible.